

# Synthesis, characterization and antibacterial activity of silver nanoparticles using *Rhazya stricta*

Adeeb Shehzad[1],*, Munibah Qureshi[1],*, Saima Jabeen[1],
Rizwan Ahmad[2], Amira H. Alabdalall[3],
Meneerah Abdulrahman Aljafary[3] and Ebtesam Al-Suhaimi[3,4]

[1] Department of Biomedical Engineering and Sciences, School of Mechanical and Manufacturing Engineering, National University of Science and Technology, Islamabad, Pakistan
[2] College of Clinical Pharmacy, Imam Abdulrahman Bin Faisal University, Dammam, Saudi Arabia
[3] Department of Biology, College of Sciences, Imam Abdulrahman Bin Faisal University, Dammam, Saudi Arabia
[4] Institute for Research and Medical Consultations, Imam Abdulrahman Bin Faisal University, Dammam, Saudi Arabia
* These authors contributed equally to this work.

Corresponding author
Ebtesam Al-Suhaimi,
ealsuhaimi@iau.edu.sa

## ABSTRACT

**Background:** Green synthesis of metallic nanoparticles has gained significant attention in the field of nanomedicine as an environment-friendly and cost-effective alternative in comparison with other physical and chemical methods. Several metals such as silver, gold, iron, titanium, zinc, magnesium and copper have been subjected to nanoformulation for a wide range of useful applications. Silver nanoparticles (AgNPs) are playing a major role in the field of nanomedicine and nanotechnology. They are widely used in diagnostics, therapeutic and pharmaceutical industries. Studies have shown potential inhibitory antimicrobial, anti-inflammatory and antiangiogenesis activities of AgNPs.
**Methods:** AgNPs have been synthesized using silver nitrate and methanolic root extract of *Rhazya stricta* that belongs to the *Apocynaceae* family. Stability and dispersion of nanoparticles were improved by adding xylitol. Synthesized nanoparticles were characterized by UV–Vis spectroscopy, scanning electron microscopy, energy dispersive spectroscopy, X-ray diffractometer and Fourier transforms infrared spectroscopy. Furthermore, the antibacterial effect of the plant extract and the nanoparticles were evaluated against gram-positive (*Bacillus subtilis*) and gram-negative (*Escherichia coli*) bacteria.
**Results:** The average size of AgNPs synthesized, was 20 nm with the spherical shape. *Rhazya stricta* based nanoparticles exhibited improved antibacterial activity against both gram-positive and negative strains.

## INTRODUCTION

Nanotechnology is a new, fascinating and fast growing revolution in the interdisciplinary fields including biomedical, therapeutics, diagnostics and materials science.

Nanotechnology involves creation and manipulation of materials at the nanoscale by altering their molecular structure (*Rajasekharreddy & Rani, 2014*). Different metals such as silver, gold, iron, titanium, zinc, magnesium and copper have been subjected to nanoformulation for a wide range of beneficial applications by adopting various physical, chemical and biological techniques (*Ponarulselvam et al., 2012*; *Painuli, Joshi & Kumar, 2018*). Green synthesis is an alternative approach for synthesizing nanoparticles (NPs) using natural resources such as microorganisms and medicinal plants as reducing agents. NPs synthesized by the green method are highly stable, environmental friendly, biocompatible, cost effective, less toxic and safe for diagnostic and therapeutic purpose (*Ponarulselvam et al., 2012*; *Rajasekharreddy & Rani, 2014*; *Verma & Mehata, 2016*).

Silver based nanomaterials play a major role in the field of nanomedicine and nanotechnology (*Logeswari, Silambarasan & Abraham, 2015*). Several studies have also proved antimicrobial activity of silver nanoparticles (AgNPs) against several pathogenic and multidrug-resistant microorganisms (*Singh et al., 2016*). Various chemical and physical techniques have been reported for the synthesis of AgNPs, however many of those techniques require costly chemicals which are also toxic (*Ahmad et al., 2017*). Therefore, the synthesis of NPs by using plant isolated compound could be an attractive strategy in the field of nanotechnology as it offers several advantages such as cost-effective, eco-friendly, rapid synthesis and high yield.

Owing to huge medicinal properties of plants, herein, it is reported green technique for the synthesis of AgNPs by using roots of *Rhazya stricta*. It is a small, glabrous, erect shrub that contains alkaloids, flavonoids and phenolic compounds (*Bukhari, Al-Otaibi & Ibhrahim, 2017*). *R. stricta* is also known as Harmal, widely distributed in Saudi Arabia and South Asia (*Bukhari, Al-Otaibi & Ibhrahim, 2017*; *Obaid et al., 2017*). Seed oil from Harmal is considered a potential rich source of d-tocopherol, a major form of vitamin E (*Nehdi et al., 2016*). It is used for treating numerous diseases such as obesity, diabetes mellitus, sore throat, cardiovascular, metabolic, neurodegenerative diseases as well as cancers (*Alagrafi et al., 2017*). It exhibits antioxidant, anticarcinogenic, antimicrobial, antidramatist, antihypertensive (*Van Beek et al., 1985*), antidepressant, anti-inflammatory, antipyretic, antifungal and herbicidal activities (*Alagrafi et al., 2017*; *Ali et al., 2000*; *Baeshen et al., 2010*; *Lantero, 2014*; *Marwat et al., 2012*; *Van Beek et al., 1985*). Previously, gold nanoparticles synthesized with extract of *R. stricta* displayed better anticancer activity against MFC-7 cell lines (*Baeshen, 2013*). In this study, we have synthesized AgNPs by using root extract of *R. stricta* plant as a reducing agent along with xylitol (sugar). The synthesized AgNPs was tested for antibacterial activity against gram-positive (*Bacillus subtilis*) and gram-negative bacteria (*Escherichia coli*).

# MATERIALS AND METHODS

## Chemicals and reagents

Silver nitrate and xylitol were purchased with $\geq 99.0\%$ purity from Sigma-Aldrich (St. Louis, MO, USA). Deionized water was used for reaction and Whatman filter papers (125 mm) were used for the purification and filtration.

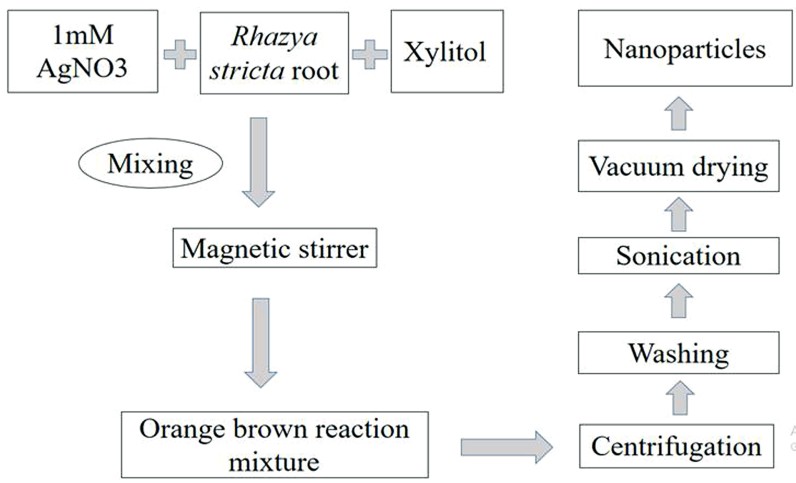

**Figure 1 Methodology for synthesis of AgNPs.** Schematic presentation of methodology for synthesis of AgNPs with root extract of *Rhazya stricta*.

## Preparation of root extract

*Rhazya stricta* was collected from southern part of Pakistan and identified in Taxonomy lab, Department of Botany at Arid Agriculture University Rawalpindi Pakistan. The voucher specimen number MAQ.2014.B313, was deposited in Department Herbarium of Botany, Arid Agriculture University Rawalpindi. The roots of the plant were washed thrice with deionized water and shade dried. Extracts were prepared by soaking 147 g of dried roots in 80% methanol at room temperature for 3 days, and then filtered using Whatman filter paper (125 mm). The solvent was evaporated using a rotary evaporator. The resulted extract after drying was 13 g.

## Biosynthesis of silver nanoparticles

For the synthesis of AgNPs, 10 g of dried extract was dissolved in 18 mL methanol and placed at 4 °C. One millimolar solution of $AgNO_3$ was prepared in deionized water. Six milliliter methanol-extracted plant roots were added drop by drop in 100 mL $AgNO_3$ (one mM) with continuous shaking. Another reaction was carried out using the same amount of extract and 100 mL of $AgNO_3$ with six mL xylitol ($10^{-2}$ M). Then both reactions were heated separately using a magnetic stirrer at 60 °C for 2 h. The change in color was the indication for synthesis of NPs (Fig. 1).

For characterization of AgNPs, the sample was centrifuged for 20 min at 12,000 rpm. The supernatant was discarded to get rid of any uncoordinated material. The sample was then washed thrice with distilled water to eliminate extra proteins or enzymes. After sonication, the sample was dried with a vacuum drier for an overnight at 50 °C.

## UV–Visible absorbance spectroscopy

Bioreduction of $Ag^+$ ions in $AgNO_3$ solution was observed by ultraviolet–visible (UV–Vis) spectroscopy (UV-2800, BMS). UV spectrums for 250, 350, 450, 550, 650 µL concentrations of plant extract in 10 mL of $AgNO_3$ solution were monitored. The UV–Vis

absorption spectra for AgNPs was observed in a range of 300–600 nm with UV-2800 spectrophotometer.

## Fourier transform infrared spectroscopy

Biomolecules responsible for reducing $Ag^+$ were analyzed by Fourier transform infrared (FTIR) spectrometer. FTIR spectroscopy is an experimental technique used for analysis of functional groups, molecular structure and chemical bonding of organic and inorganic samples, by producing an infrared absorption spectrum (Perklin Elmer Spectrum 100) (*Verma & Mehata, 2016*).

## X-ray diffraction

The structural characterization and crystalline nature of AgNPs were determined by X-ray diffractometer (XRD; D8 Advance; Bruker, Billerica, MA, USA) for the vacuumed dried AgNPs (*Ashraf et al., 2016*).

## Scanning electron microscopy

The size and morphology of AgNPs were investigated using JEOL-6490A-JSM scanning electron microscope (SEM). Elemental composition of AgNPs was determined by energy dispersive spectroscopy.

## Antibacterial activity

The antibacterial activity of the extract and AgNPs was evaluated against *E. coli*, and *B. subtilis* using disk diffusion method (*Hu et al., 2017*). Disks were soaked with distilled water (negative control), standard antibiotic cefipime (positive control) with the trade name of maxipime (glaxosmithkline), *R. stricta* extract and the AgNPs solution with and without xylitol addition separately. A total of 50 µL concentration of AgNPs was used to ensure identification of antibacterial activity. Then the plates were incubated for 24–48 h at 37 °C and zone of inhibition was measured for both bacterial strains.

# RESULTS

## UV–Vis spectroscopy

Synthesis of NPs started after the addition of root extract of *R. stricta* into an aqueous solution of $AgNO_3$. The change in color from pale yellow to orange brown after 2 h heating (60 °C) on magnetic stirrer indicated the formation of reduced silver (Fig. 2A).

To examine further AgNPs in aqueous suspensions, UV–Vis spectroscopy was used. As shown in the Fig. 2B, the absorption peaks of AgNPs synthesized by *R. stricta* root extract were detected at 380, 392, 402, 414 and 427 nm wavelength. The increasing concentration of plant extracts also increased the broadening of the peak indicating the increased size of NPs. Previous studies indicated that the spherical AgNPs have absorption bands at around 400 nm in the UV–Visible spectra (*Nasir, Mohammed & Samir, 2016*). These results indicate that the spherical AgNPs have absorption bands at around 402 nm in the UV–Visible spectra. Thus, it can be reported that particle size of AgNPs is highly subjective to the concentration of root extract.

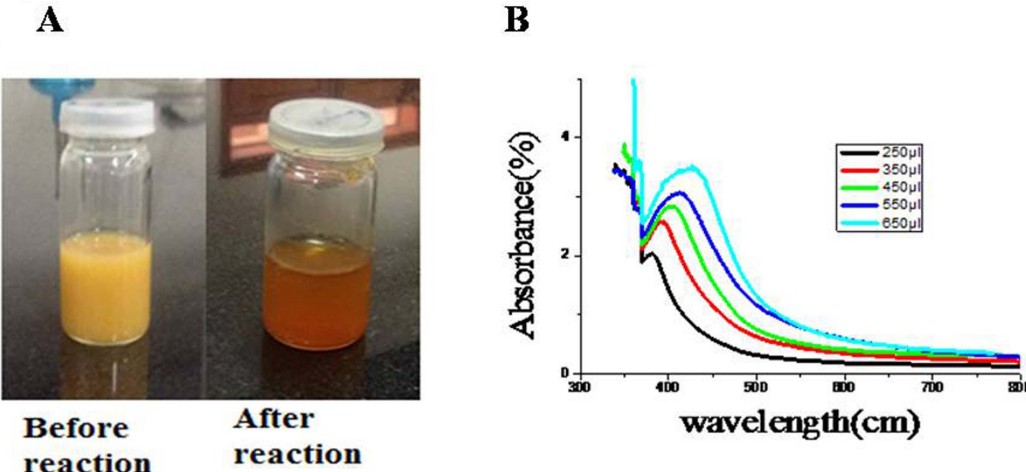

**Figure 2 Charecterization of AgNPs.** (A) Change in color after synthesis of AgNPs. (B) Concentration-dependent UV–Vis Spectra of AgNPs of *R. stricta*. Photo credit: Saima Jabeen.

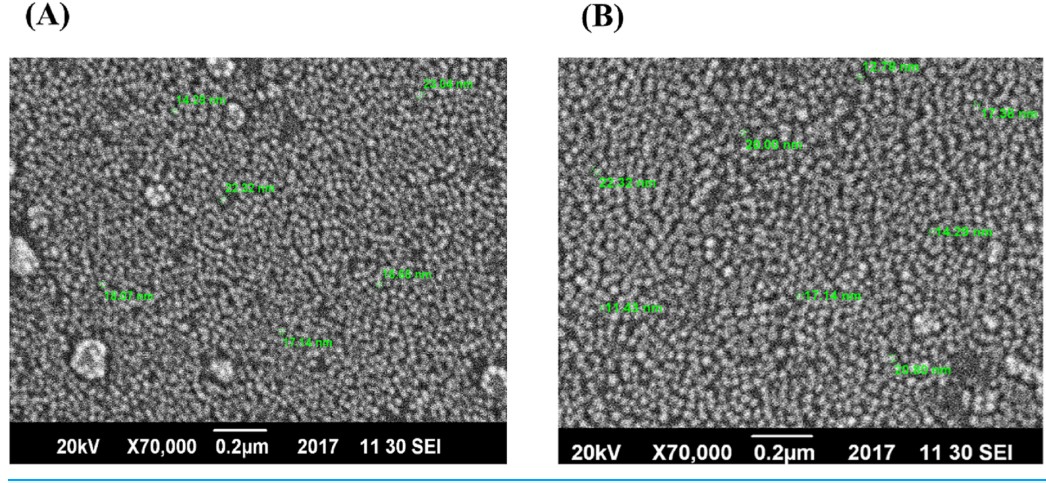

**Figure 3 SEM images of AgNPs.** (A) AgNPs of plant extract only. (B) AgNPs of plant extract with xylitol.

## Scanning electron microscopy

For further confirmation, the size and shape of AgNPs, SEM was performed. Figures 3A and 3B depict the formation of spherical shape NPs. It was noticed that AgNPs synthesized only with plant extract have shown more agglomeration (Fig. 3A), while those synthesized with the addition of xylitol were not agglomerated, well dispersed and with average particle size of 20 nm.

## Elemental analysis

The AgNPs synthesized using *R. stricta* were also characterized by EDX analysis, which further confirmed the reduction of silver ion. The Optical absorption peak was recorded almost at three kV, which is obvious for silver nanocrystals absorption because of surface Plasmon resonance (Fig. 4). The EDX spectrum has shown silver peak along

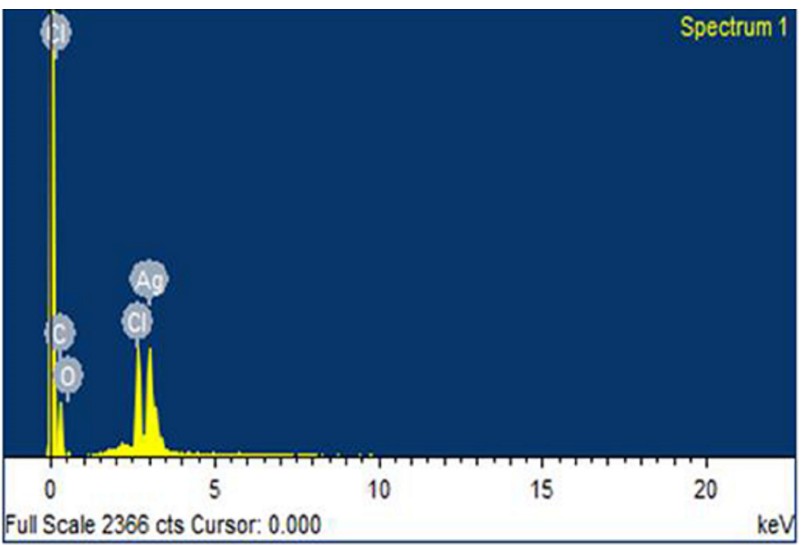

**Figure 4 EDX spectrum.** Energy dispersive X-ray image of AgNPs synthesized using *Rhazya stricta*.

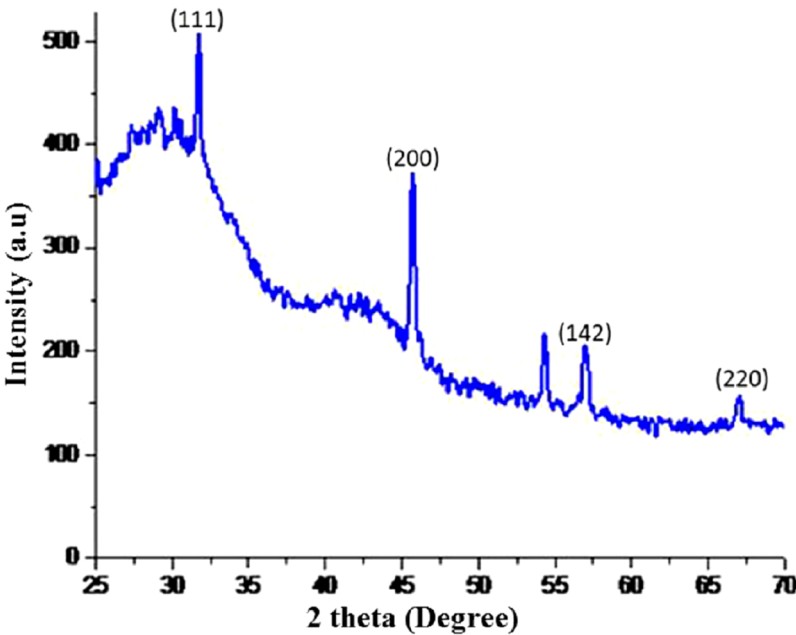

**Figure 5 XRD patterens.** X-ray diffraction (XRD) image of the AgNPs synthesized using *Rhazya stricta*.

with oxygen and chlorine peaks. The chlorine and oxygen peaks were may be attributed to glass biomolecules that were present on the surface of AgNPs or to chlorine on the glass slides that were used for sample preparation (Fig. 4).

## X-ray diffraction

The dry powders of AgNPs synthesized with root extract of *R. stricta* were subjected to XRD analysis. XRD patterns of synthesized AgNPs are shown in Fig. 5. The synthesized

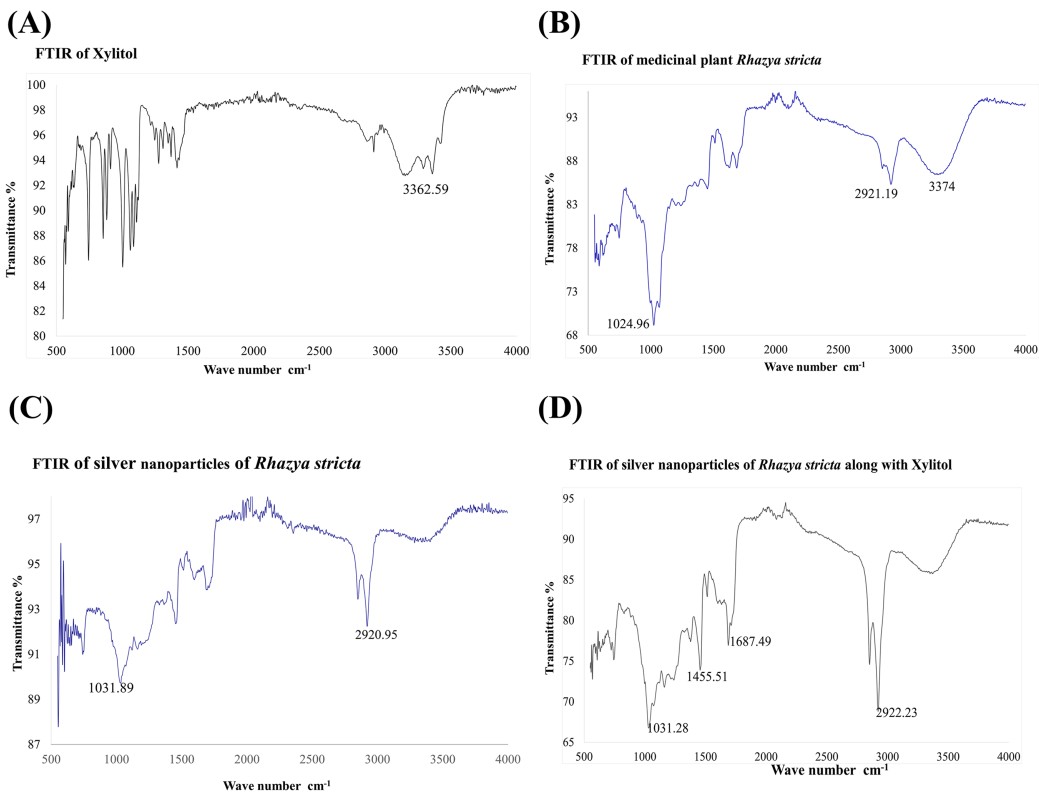

**Figure 6 FTIR spectrum.** (A) FTIR of Xylitol. (B) FTIR of medicinal plant *R. stricta*. (C) FTIR of AgNPs using *Rhazya stricta*. (D) FTIR of AgNPs using *Rhazya stricta* along with xylitol.

NPs were of crystalline nature. The XRD pattern has shown the intense five peaks of NPs at 31.99, 45.5, 54.85, 57.52 and 67.24, were indexed to face-centered cubic (fcc) structures of AgNPs. The XRD peaks revealed that AgNPs formed using *R. stricta* root extract were crystalline and spherical in shape. The average nanosize was confirmed by applying Debye–Scherrer formula $D = K \lambda/\beta \cos\theta$ which was found to be 20 nm.

## Fourier transform infrared spectroscopy

Fourier transform infrared spectroscopy was used to investigate. FTIR analysis was done to identify biomolecules that are responsible for $Ag^+$ ions reduction and capping of biologically reduced AgNPs synthesized by *R. stricta* root extract along with xylitol. Figure 6A shows the FTIR spectrum of xylitol displayed bands at 3,362.59 (OH of alcohol) and 1,417.88 (C–H bending of alkane). Figure 6B shows the FTIR spectrum of root extract of *R. stricta* plant that is showing peaks at 724, 1,024, 1,455, 1,687, 2,123, 2,921 and 3,374 cm$^{-1}$. A peak at 3,374 cm$^{-1}$ indicated stretching of the N–H bond of amino groups and presence of attached hydroxyl (–OH) group. The absorption peaks at 2,921 cm$^{-1}$ could be due to stretching of –CH functional groups. The peak at 1,687 cm$^{-1}$ corresponds to C–O stretching in the carboxyl attached to the amide linkage in amide I. The absorption band at 1,455 indicated the presence of C–N stretching in amide. The two bands present at 1,024 and 742 cm$^{-1}$ could be indicative of the –O– stretching vibrations of

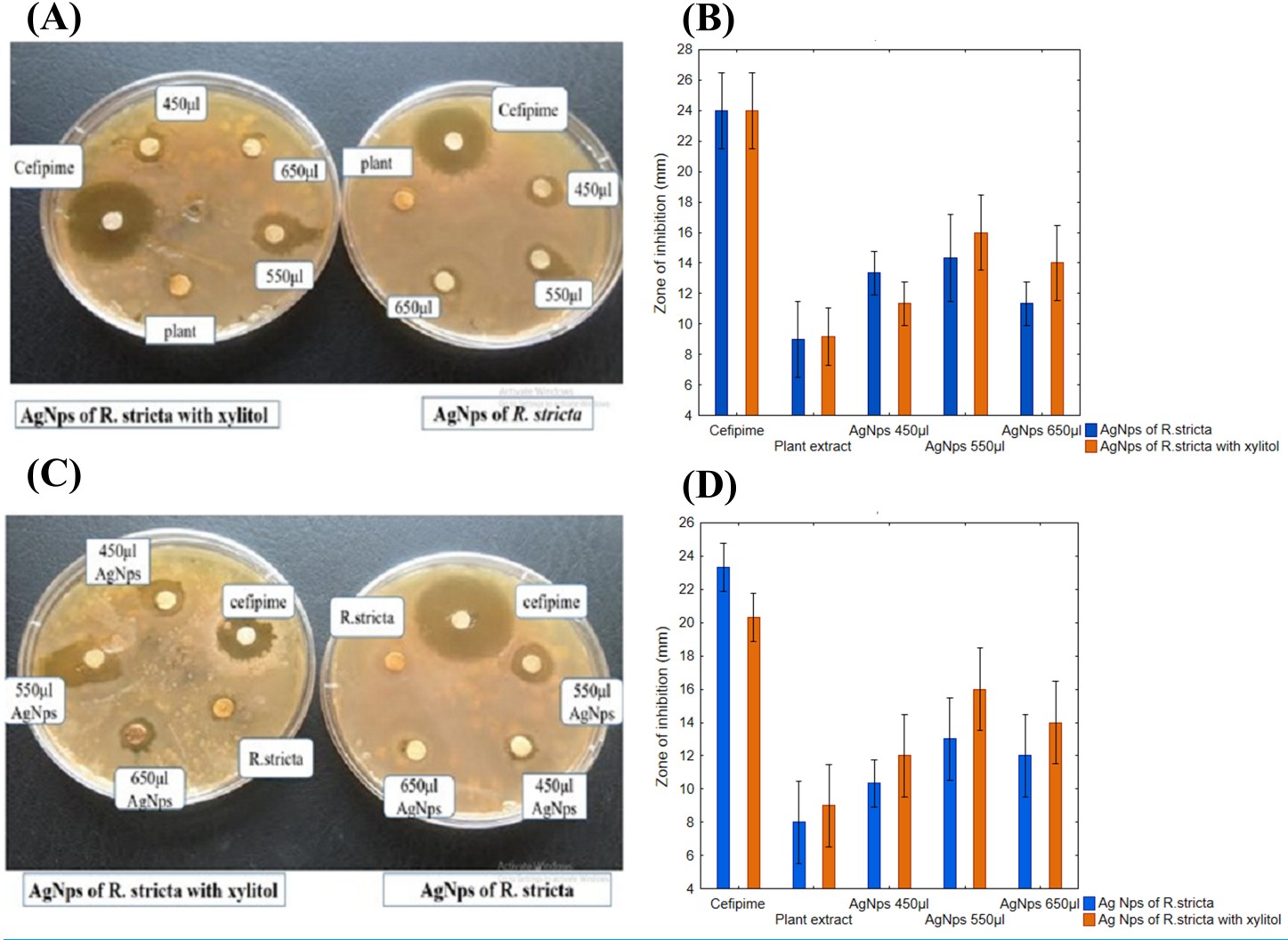

**Figure 7 Concentration-dependent antimicrobial activity of Ag NPs.** (A) Concentration-dependent effect of *B. subtilis*. (B) Graphical presentation for concentration-dependent effect of *B. subtilis*. (C) Concentration-dependent effect of *E. coli*. (D) Graphical presentation for concentration-dependent effect of *E. coli*.   

aromatic and aliphatic amines (*Gou et al., 2015*). Figures 6C and 6D show FTIR spectra for AgNPs synthesized using only *R. stricta* plant and along with xylitol.

## Antimicrobial activity of AgNPs

Plant mediated synthesis of AgNPs exhibited tremendous antibacterial activity against various bacterial strains. For this purpose, the antibacterial activity of AgNPs was examined with various concentrations of root extract against *B. subtilis* for gram-positive and *E. coli* for gram-negative strains. As shown in the Figs. 7A and 7C, three different concentrations (450, 550 and 650 µL) of plant extract in 10 mL of AgNO$_3$ were used to test the antibacterial activity. When concentration was increased from 450 to 550 µL, the zone of inhibition also increased, however, at 650 µL, zone of inhibition was observed smaller than 550 µL (Figs. 7A and 7C). The positive control 10 mM of cefipime exhibited maximum zone of inhibition, whereas distilled water, the negative control shown

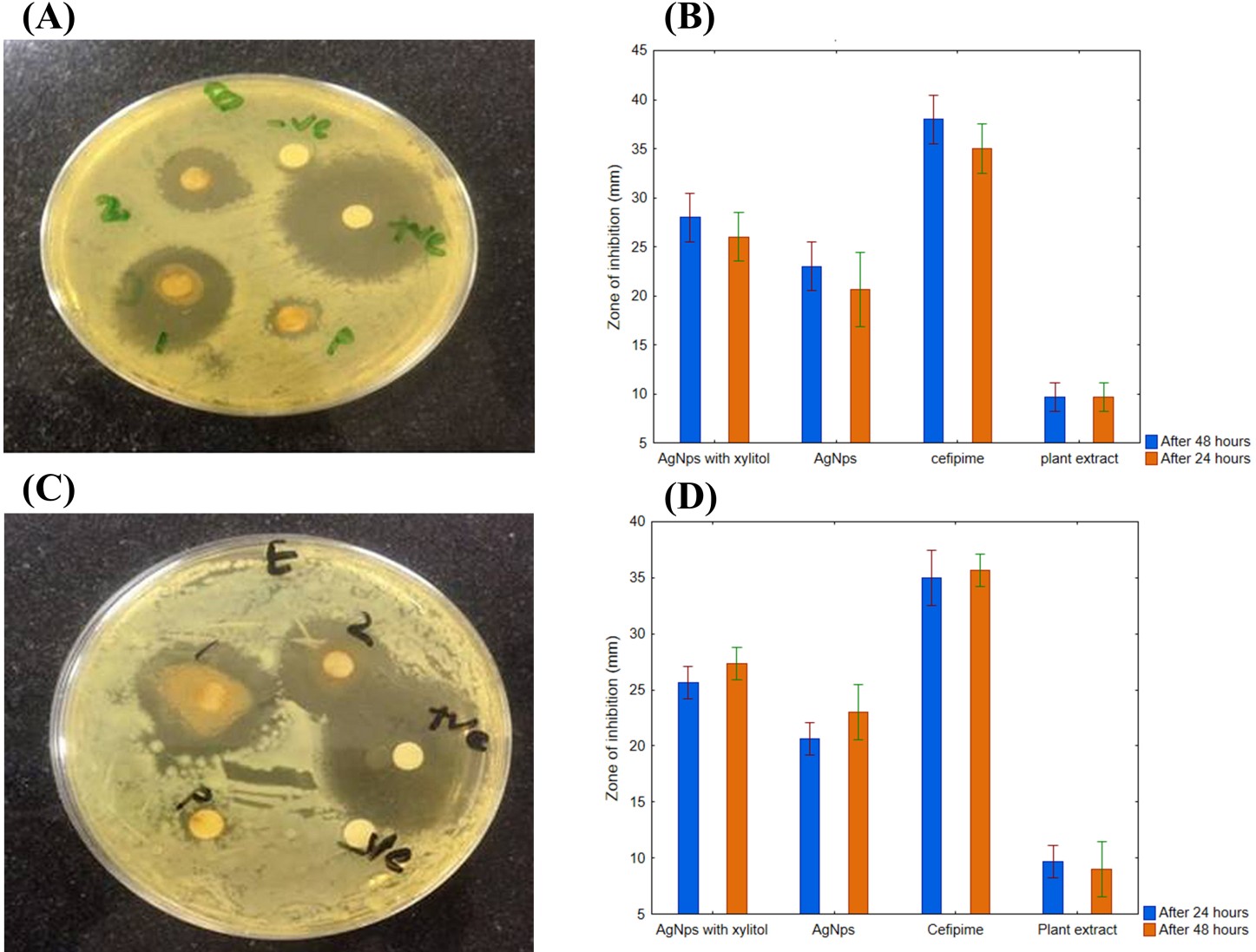

**Figure 8 Time-dependent antimicrobial activity of AgNPs.** (A) Time-dependent effect of *B. subtilis*. (B) Graphical presentation for time-dependent effect of *B. subtilis*. (C) Time-dependent effect of *E. coli*. (D) Graphical presentation for time-dependent effect of *E. coli*.

no inhibition on both strains (Figs. 7B and 7D). The AgNPs synthesized by *R. stricta* root extract along with xylitol displayed better antibacterial activity as compared to plant extract only, because xylitol increased the stability and decreased the agglomeration of synthesized AgNPs. Thus, uniform distribution of NPs enhanced the antibacterial activity of synthesized AgNPs with root extract of *Rhyza stricta*.

## Time-dependent effect of AgNPs

Once the concentration-dependent effect was confirmed, the effect of AgNPs synthesized using *R. stricta* root extract was examined for its time-dependent activity. The 550 μL of AgNPs with *R. stricta* root extract was tested further with and without xylitol against *B. subtilis* (gram-positive) and *E. coli* (gram-negative). As shown in Figs. 8A and 8B,
**(A)**

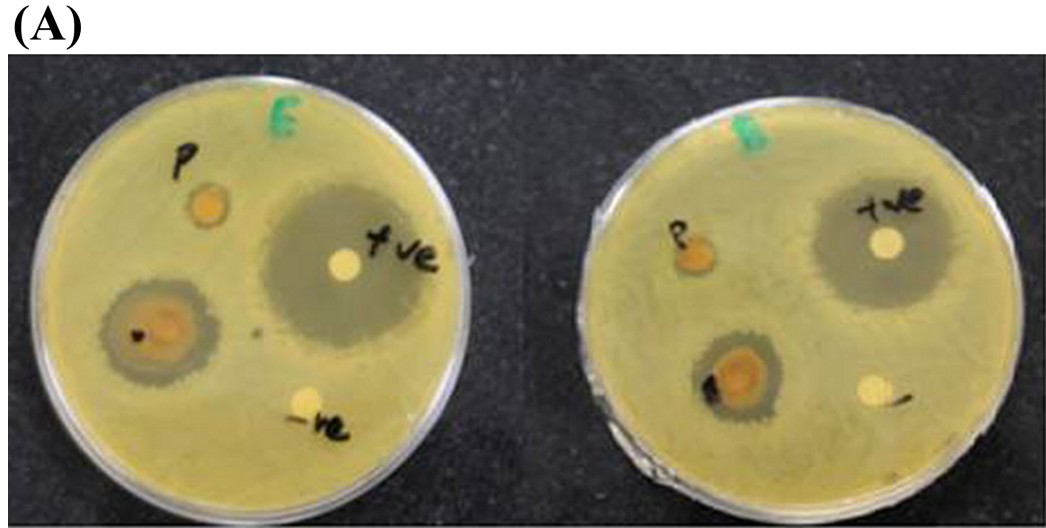

**(B)**

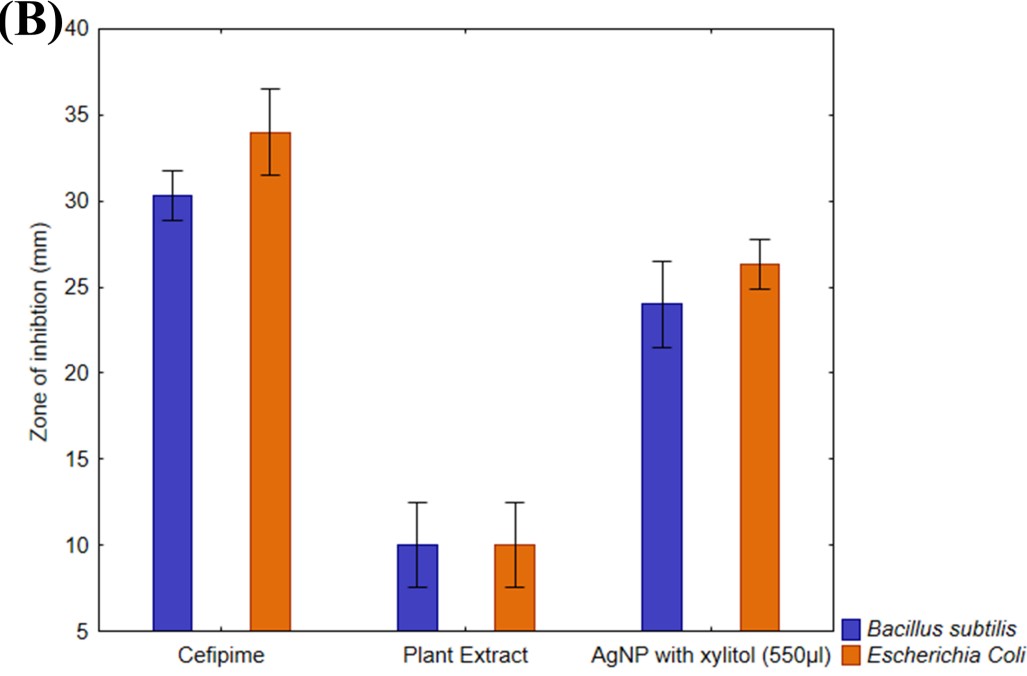

Figure 9 **Comparative effect of AgNPs against *E. coli* and *B. subtilis.*** (A) Difference between effect of silver nanoparticles of *R. stricta* along with xylitol on *E. coli* and *B. subtilis*. (B) Graphical presentation for comparative effect of AgNPs against *E. coli* and *B. subtilis*.

AgNPs with *R. stricta* root extract and xylitol have shown an increased zone of inhibition as compared to their individual zone of inhibition. These results confirm the enhanced antibacterial effect of AgNPs synthesized with *R. stricta* root extract and xylitol.

## Differential effect of AgNPs against gram-negative and positive pathogens

The synthesized AgNPs had more inhibition effect on *E. coli* as compared to *B. subtilis* as shown in Figs. 9A and 9B. These results further confirmed that AgNPs synthesized

with *R. stricta* root extract have better bactericidal effects against gram-negative strains as compared to gram-positive strains.

## DISCUSSION

Green synthesis of AgNPs could significantly overcome the problem of using chemical agents that cause various adverse effects. Therefore, green synthesis of NPs is environment-friendly approach (*Gan & Li, 2012*; *Wang et al., 2017*). *R. stricta* plant roots possess antibacterial activity, that is, considerably enhanced after incorporation with NPs. The characterization techniques advocated the stability of the synthesized AgNPs. Experimentally, Xylitol has not the ability to reduce silver ions, but it is possibly involved in the capping of the synthesized AgNPs by electrostatic attraction or binding with the protein groups present in the extract through hydrogen bonding and enhanced the silver nanoparticle1's stability. It also protected the NPs from agglomeration (*Kaviya, Santhanalakshmi & Viswanathan, 2011*). Synthesis of the AgNPs starts with the change in color from pale yellow to orange brown upon the addition of *R. stricta* root extract (Fig. 2A). It has been reported that excitation of surface Plasmon resonance is responsible for yellowish brown color of AgNPs in aqueous solution (*Kaviya, Santhanalakshmi & Viswanathan, 2011*). This surface Plasmon vibration confirmed the biotransformation from an ionic form to silver particles (*Huang et al., 2007*). Upon addition of *R. stricta* root extract in the aqueous solution of $AgNO_3$ and xylitol, the color changed from pale yellow to orange brown due to reduction of silver ion (Fig. 2A). It was also noticed that increase in the concentration of root extract in the synthesis reaction, increases the particle size. Further increase in plant extract concentration shown noisy and broadened peak at UV–Vis spectrum. Current XRD results of synthesized AgNPs with root extract of *R. stricta* confirmed the structure of AgNPs as fcc with crystalline nature (*Li et al., 2012*). The EDX spectrum has shown clear silver signal along with weak oxygen and chlorine peak, which might be due to the interaction and binding of biomolecules to the surface of AgNPs (*Kalainila et al., 2014*).

Fourier transform infrared analysis was performed to confirm that the biocompounds involved in the reduction of ionic and capping of the reduced AgNPs synthesized by *R. stricta* root extract along with xylitol. Figure 6A shows FTIR spectrum of xylitol, it gave bands at 3,362.59 (OH of alcohol) and 1,417.88 (C–H bending of alkane). Figure 6B represents the FTIR spectrum of root extract of *R. stricta* plant; it gave peaks at 724, 1,024, 1,455, 1,687, 2,123, 2,921 and 3,374 $cm^{-1}$. These peaks are expected to be related with amide I shift due to carbonyl stretch in proteins (1,687 $cm^{-1}$), –C–H– stretch of amine (2,921 $cm^{-1}$), C–N stretching vibration of amine (1,455 $cm^{-1}$) and –O– stretching vibrations of aliphatic and aromatic amines (1,024 and 742 $cm^{-1}$), respectively (*Gou et al., 2015*; *Wang et al., 2017*). A peak at 3,374 $cm^{-1}$ indicated the stretching of the N–H bond of amino groups and shown the presence of bonded hydroxyl (–OH) group (*Gou et al., 2015*; *Hu et al., 2017*; *Wang et al., 2017*). After the formation of AgNPs peaks were shifted to little higher wavenumber as compared to plant extract as indicated by Fig. 6C. Xylitol did not reduce the silver ions in the solution, but it was used to cap/coat the synthesized

AgNPs through electrostatic attraction or hydrogen bonding with protein groups that further aided in more stability and better dispersion of NPs Fig. 6D. FTIR spectrum of the *R. stricta* roots extract confirmed the presence of carboxyl (–C=O), hydroxyl (–OH) and amine (N–H) groups of *R. stricta* roots extract and further revealed their involvement in the reduction of silver ion to metallic AgNPs (*Nasir, Mohammed & Samir, 2016*). Plant extract contains proteins that cap the AgNPs through carboxyl or amino group present in protein (*Kaviya, Santhanalakshmi & Viswanathan, 2011*). It is believed that proteins of the root extract of *R. stricta* could bind to AgNPs through either free amino or carboxyl groups (*Nasir, Mohammed & Samir, 2016*). The presence of the residual plant extract in the sample is due to the similarity of the spectra with little marginal shifts in peak position (*Ikram & Ahmed, 2015*). Thus, these results have shown that the functional groups of biocompounds present in the root extract of *R. stricta* have key role in the reduction as well as capping of AgNPs.

Previous studies have also reported the antibacterial activity of AgNPs against gram-negative and gram-positive strains of bacteria (*Singh et al., 2008*). It is well recognized that the antibacterial effect of AgNPs is size and dose dependent. The results were supported by previously reported studies (*Sana & Dogiparthi, 2018*), which have shown that AgNPs have more potential against gram-negative bacteria then gram-positive bacteria (Figs. 7A and 7C). Increase in the size of NPs reduced their antibacterial activity (*Kaviya, Santhanalakshmi & Viswanathan, 2011*). *R. stricta*-AgNPs exhibited strong antioxidant and antibacterial effects It is also concluded that the antibacterial activity of AgNPs depends on the concentration of root extract and quantity of AgNPs in the medium (*Rauf et al., 2016*).

## CONCLUSION

Silver nanoparticles synthesized using *R. stricta* root extract along with xylitol have shown potential antimicrobial activity because they were mono dispersed, stable and have less aggregation. The plant acted as a reducing agent in the synthesis of AgNPs and xylitol enhanced the dispersion of NPs. NPs of a smaller size displayed better antibacterial activity. Antibacterial activity of *R. stricta* plant root was enhanced when it was incorporated with AgNPs synthesis. Thus, synthesized AgNPs increases the therapeutic efficacy, strengthen the medicinal values of *R. stricta* and provides a potent source of an antimicrobial agent against gram-positive and gram-negative organisms. Future studies are needed to uncover the underlying mechanism of bacteriostatic or bactericidal effects of green synthesized AgNPs.

### Funding

The authors received no funding for this work.

### Competing Interests

The authors declare that they have no competing interests.
## Author Contributions

- Adeeb Shehzad conceived and designed the experiments, performed the experiments, analyzed the data, prepared figures and/or tables, authored or reviewed drafts of the paper, approved the final draft.
- Munibah Qureshi conceived and designed the experiments, analyzed the data, contributed reagents/materials/analysis tools, prepared figures and/or tables, authored or reviewed drafts of the paper.
- Saima Jabeen conceived and designed the experiments, performed the experiments, analyzed the data, prepared figures and/or tables.
- Rizwan Ahmad performed the experiments, contributed reagents/materials/analysis tools, prepared figures and/or tables.
- Amira H. Alabdalall contributed reagents/materials/analysis tools, prepared figures and/or tables.
- Meneerah Abdulrahman Aljafary contributed reagents/materials/analysis tools, prepared figures and/or tables, authored or reviewed drafts of the paper.
- Ebtesam Al-Suhaimi conceived and designed the experiments, analyzed the data, approved the final draft.

## Data Availability

The raw data are provided in the Supplemental Files.

## Supplemental Information

Supplemental information for this article can be found online at http://dx.doi.org/10.7717/peerj.6086#supplemental-information.

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
