# Peer review of "Synthesis, characterization and antibacterial activity of silver nanoparticles using Rhazya stricta"

_PeerJ, doi:10.7717/peerj.6086_

## Round 0.1 · original submission · Major Revisions

Please address in detail the comments of the 3 reviewers. Clearly mark your changes in a tracked changes document and justify each edit in a formal rebuttal letter.

Reviewer 1 ·

Basic reporting

no comments

Experimental design

no comments

Validity of the findings

no comment

Additional comments

1. Overall, the manuscript is concise, clear and well written. Introduction provides a good, comprehensive background of the topic that makes the reader a clear understanding of the work. However, the author needs to give some background about previous work regarding synthesis and application of any other metallic nanoparticles using the same plant, Rhazya stricta(if studied), with appropriate references, which may further potentiate the authors work.
2. The methods and experimental techniques used in the study for synthesis and characterizations are generally appropriate for the study.
3. The results obtained are coherent with the objectives defined. However, the author should demonstrate more clearly that the results obtained in current study are consistent with the results found in previous studies (mainly for synthesis section). Discussion regarding other metallic nanoparticles (if any) using same plant in previous studies will be appreciated.
4. References need to be formatted. Within text citation should be according to journal guidelines. Either the abbreviation or full name of the journal should be mentioned in complete references throughout. Moreover, some references include bold text that must be corrected.
The manuscript is recommended for publication if author responds to above comments properly as these issues cannot be overlooked.

Reviewer 2 ·

Basic reporting

Peer J: Manuscript 28748v1
Manuscript title Synthesis, characterization and antibacterial activity of silver nanoparticles using Rhazya stricta by green synthesis.
The manuscript reports on "Synthesis, characterization and antibacterial activity of silver nanoparticles using Rhazya stricta by green synthesis" looks very interesting and fascinating and will help for further studies related to activities Rhazya stricta by green synthesis. The article looks suitable and I suggest for the acceptance. The idea looks novel and supported very well by experimental results, however, I suggest following minor comments before publication.
1. Generally, the abstract and the explanatory part of the paper are well presented, however;
The background looks too long please shorten it by modifying other sentence or remove the last sentence. “They are widely used in diagnostics, therapeutic and pharmaceutical industries. Studies have shown potential inhibitory antimicrobial, anti-inflammatory and anti-angiogenesis activities of AgNPs”.
2. The last sentence of the result should be like a conclusion and should be more concise. Please remove the future prospect from the result section. “AgNPs can be used for enhancing the efficacy and retention time of phytochemicals, however, in-vivo, preclinical and clinical studies are required to completely address its therapeutic potential in various diseases”.
3. Please remove the repeated words from the following consecutive sentences.
Currently, green synthesis of NPs is gaining significant attention over the physical and chemical methods, because it exhibits greater catalytic activity and reduces the use of costly and lethal chemicals (Ponarulselvam et al., 2012; Verma&Mehata, 2016). NPs synthesized by the green method are highly stable, environment-friendly, biocompatible, cost-effective, less toxic and safe for the diagnostic and therapeutic purpose (Ponarulselvam et al., 2012; Rajasekharreddy& Rani, 2014; Verma&Mehata, 2016).
4. Please put a space between numbers and units. Sucha 147g etc
5. Please look for the references and make a uniform style.

Experimental design

Experiments are designed well and materials and methods are described with sufficient detail & information are properly written.

Validity of the findings

The data are meaningful, and backgrounds research literature are clearly described. The conclusion is well drawn for future perspective.

Additional comments

The data are meaningful, and backgrounds research literature are clearly described. The conclusion is well drawn for future perspective.

Reviewer 3 ·

Basic reporting

Check the general comments

Experimental design

Check the general comments

Validity of the findings

Check the general comments

Additional comments

The manuscript entitled “Synthesis, characterization and antibacterial activity of silver nanoparticles using Rhazya stricta” seems to be interesting. The manuscript is of acceptable quality after minor revisions.
The authors should address the following points before the manuscript is accepted
Generally, the abstract and the explanatory part of the paper are well presented, however;
1. Currently, the area of nanotechnology is a fresh front line of science and technology that manipulates matter at the nanometer scale (1-100nm change to 1-100 nm) so what is the actual size of the synthesized nanoparticles and how it has been confirmed in this study.
2. To date, a lot of literature has reported on biological synthesis of Silver nanoparticles using plants as well as microorganisms including bacteria and fungi, so what was the condition of the plant that has been used in the synthesis of NPs.
3. The author needs to explain the Fourier Transform Infrared Spectroscopy (FTIR) in Materials and Methods.
4. The author needs to write the trade name and company name as well as dose for cefipime.
5. How the author choose the concentration for the time dependent effect in this study.
6. The last sentence should be corrected ‘Future studies are needed to uncover the underlying mechanism of bacteriostatic or bactericidal effects to green synthesized AgNPs’.
7. In references there is dots before the last name, it is the format of the references or mistakes, author should take care for the uniformity of references.

---

## Round 0.2 · Major Revisions

Reviewer 4 still has remaining comments for you. Please respond to them in detail.

Reviewer 1 ·

Basic reporting

Manuscript ha been well revised.

Experimental design

Experimental part is ok now

Validity of the findings

Ok

Additional comments

Acceptable after revision

Reviewer 2 ·

Basic reporting

The article is clear with proper literature reviews of the background knowledge, figures and tables are properly arranged, and results are well discussed.

Experimental design

Experimental section is properly arranged. changes are made according to suggestion.

Validity of the findings

Finding are discussed properly.

Reviewer 3 ·

Basic reporting

accept

Experimental design

ok

Validity of the findings

ok

Additional comments

accept

Reviewer 4 ·

Basic reporting

The basic reporting looks good however leaving ample of scope for improvements.

Experimental design

The Experiments are quite pertinent to the topic.

Validity of the findings

The findings seems to be good however the element of novelty is missing. Mere choosing new plant species for the synthesis of silver nanoparticles and reporting antibacterial activity is not worthy of publication.

Additional comments

In the present manuscript the authors have reported synthesis of silver nanoparticles using plant extract of Rhazya stricta and later employed them for the antibacterial studies. Though the authors have satisfactorily addressed the comments of the previous reviewers, still some concerns need to be addressed.
1. Several papers have been published on the green synthesis of silver nanoparticles, their characterization and their biological effectiveness studied in in vitro models or their dye degradation potential. They generally differ in the choice of plant extract used in the nanoparticle preparation, the bacterial species for antibacterial studies or the dye that is being shown to degrade using nanoparticles. Though the work reported in the manuscript appears technically sound and well written, in my opinion, it does not appear to offer anything more useful than that in the other similar papers except the use of new plant extract. There is an urgent need to move ahead of just reporting another plant/bacteria/ yeast for green synthesis of silver nanoparticles. There should be studies that could report the in vivo effects of these nanoparticles. A simple study using disc diffusion is not sufficient at this time point. Here, it would be worth publishing if the authors could include some studies for deciphering the mechanism by which the silver nanoparticles reported here are exerting the antibacterial effect. Also a look inside the bacterial cells for the morphological or anatomical changes due to exposure to silver nanoparticles would be worth publishing.
2. Figure 3- the size values shown in the figure are not clear, hence it’s difficult to conclude about the size of nanoparticles.
3. Figure 5- the peaks corresponding to fcc of silver should be labeled.
4. Figure 6-Significant peaks should be labeled in the FT IR spectra and if possible they should be overlayed to facilitate understanding.
5. Figure 8,B,D- all the error bars look same, it’s not clear how many times the experiment was done. The label of y-axis is also missing.
6. Figure 9-B again the y-axis label is missing and the no. of experiments done is not clear or mentioned.
7. Please refer these articles in order to improve the presentation of the figures.
a. Synthesis and in vitro antineoplastic evaluation of silver nanoparticles mediated by Agrimoniae herba extract. Qu D, Sun W, Chen Y, Zhou J, Liu C. Int J Nanomedicine. 2014 Apr 15;9:1871-82. doi: 10.2147/IJN.S58732. eCollection 2014.
b. Green synthesis of silver nanoparticles using Prosopis juliflora bark extract: reaction optimization, antimicrobial and catalytic activities. Arya G, Kumari RM, Gupta N, Kumar A, Chandra R, Nimesh S. Artif Cells Nanomed Biotechnol. 2018 Aug;46(5):985-993. doi: 10.1080/21691401.2017.1354302. Epub 2017 Jul 18.

---

## Round 0.3 · Minor Revisions

I do not believe that you need to do more experiments, as requested by Reviewer 4.

Please revise to provide a clearer description of the FTIR spectra in lines 258-265 (for example, you do not discuss the spectra of the AgNP).

The electron microscopy photos have very poor resolution. Please submit an improved version, if you have high-resolution images.

Minor corrections:

Line 274-277 Please check the reference.
Figure captions: Line 414
417 420 Please check Escherichia Coli it should be Escherichia coli
Line 410 Please correct R. stricta.

Reviewer 4 ·

Basic reporting

No comment

Experimental design

No comment

Validity of the findings

No comment

Additional comments

Though the manuscript has been improved as suggested but the first point remains unaddressed. Even the authors have agreed that this study is incomplete and they are pursuing the experiments in the lab and would be interested in publishing at some later point, with those results.
As per my understanding the present manuscript does not seem to have sufficient novelty for publication and hence I am not in the favor of acceptance for publication. I would encourage the authors to submit the manuscript only after incorporating the results of experiments that deal with deciphering the mechanism of antibacterial effect of the AgNPs.

---

## Round 0.4 · accepted · Accept

No more revision is required.

#